# The Adipokinetic Hormone (AKH) and the Adipokinetic Hormone/Corazonin-Related Peptide (ACP) Signalling Systems of the Yellow Fever Mosquito *Aedes aegypti*: Chemical Models of Binding

**DOI:** 10.3390/biom14030313

**Published:** 2024-03-06

**Authors:** Graham E. Jackson, Marc-Antoine Sani, Heather G. Marco, Frances Separovic, Gerd Gäde

**Affiliations:** 1Department of Chemistry, University of Cape Town, Private Bag, Rondebosch 7701, South Africa; graham.jackson@uct.ac.za; 2Bio21 Institute, University of Melbourne, Melbourne, VIC 3010, Australia; fs@unimelb.edu.au; 3Department of Biological Sciences, University of Cape Town, Private Bag, Rondebosch 7701, South Africa; heather.marco@uct.ac.za (H.G.M.); gerd.gade@uct.ac.za (G.G.); 4School of Chemistry, University of Melbourne, Melbourne, VIC 3010, Australia

**Keywords:** *Aedes aegypti*, adipokinetic hormone, corazonin hormone, molecular modelling, yellow fever mosquito, peptide signalling

## Abstract

Neuropeptides are the main regulators of physiological, developmental, and behavioural processes in insects. Three insect neuropeptide systems, the adipokinetic hormone (AKH), corazonin (Crz), and adipokinetic hormone/corazonin-related peptide (ACP), and their cognate receptors, are related to the vertebrate gonadotropin (GnRH) system and form the GnRH superfamily of peptides. In the current study, the two signalling systems, AKH and ACP, of the yellow fever mosquito, *Aedes aegypti*, were comparatively investigated with respect to ligand binding to their respective receptors. To achieve this, the solution structure of the hormones was determined by nuclear magnetic resonance distance restraint methodology. Atomic-scale models of the two G protein-coupled receptors were constructed with the help of homology modelling. Thereafter, the binding sites of the receptors were identified by blind docking of the ligands to the receptors, and models were derived for each hormone system showing how the ligands are bound to their receptors. Lastly, the two models were validated by comparing the computational results with experimentally derived data available from the literature. This mostly resulted in an acceptable agreement, proving the models to be largely correct and usable. The identification of an antagonist versus a true agonist may, however, require additional testing. The computational data also explains the exclusivity of the two systems that bind only the cognate ligand. This study forms the basis for further drug discovery studies.

## 1. Introduction

Three neuropeptide signalling systems exist in invertebrates, where the mature peptides and their cognate G protein-coupled receptors (GPCRs) are, respectively, structurally similar. Collectively, these signalling systems are related to the vertebrate gonadotropin-releasing hormone (GnRH) system, giving rise to the concept of a large peptide superfamily [1,2,3,4]. The aforementioned invertebrate neuropeptide systems are the adipokinetic hormone (AKH)/red pigment-concentrating hormone (RPCH) family, the corazonin (Crz) family, and the structurally intermediate family known as the adipokinetic hormone/corazonin-related peptide (ACP) family (see a recent review by Marco et al. [5]). In the current study, we focus on members of the GnRH superfamily in the medically relevant pest insect, the mosquito *Aedes agypti*, hence, a description of the GnRH superfamily will be restricted here to insects.

AKHs are primarily or exclusively produced in neurosecretory cells of the corpus cardiacum (CC), and a major function is the mobilisation of energy reserves stored in the fat body, thus providing an increase in the concentration of diacylglycerols, trehalose, or proline in the haemolymph for locomotory active phases. Accordingly, the AKH receptor (AKHR) transcripts are predominantly found in fat body tissue, where the AKH activates the enzymes glycogen phosphorylase and triacylglycerol lipase, respectively [6,7,8]. As with most endocrine regulatory peptides, additional functions, such as inhibition of anabolic processes (protein and lipid syntheses), involvement in oxidative stress reactions, and egg production inter alia are known, making AKH a truly pleiotropic hormone [9,10].

Crz is mainly synthesised in neuroendocrine cells of the pars lateralis of the protocerebrum and released into circulation via the CC [11]. Although originally described as a potent cardio-stimulatory peptide [12], Crz does not generally fulfil this role but is known for other functions, such as involvement in the release of pre-ecdysis and ecdysis-triggering hormones, reduction of silk spinning rates in the silk moth, pigmentation events (darkening) of the epidermis in locusts as they transition to the gregarious phase, and regulation of caste identity in an ant species [13,14,15,16].

The functional role of ACP is less clear. Previous studies had not found a clear-cut function for this peptide in *Anopheles gambiae* and *Rhodnius prolixus* [17,18] until work by Zhou et al. [19] claimed that ACP in male crickets (*Gryllus bimaculatus*) regulates the concentration of carbohydrates and lipids in the haemolymph. This, however, could not be verified in independent experiments (H.G. Marco and G. Gäde, unpublished observations). In 2021, Hou et al. [20] reported the involvement of ACP in the regulation of lipid use during long-distance flight in *Locusta migratoria*, specifically in the oxidation and transport of fatty acids in the flight muscles. Most recently, a surprising sex-specific role of ACP in adult *A. aegypti* was put forward by Afifi et al. [21]: in adult female mosquitos, abdominal glycogen content decreased upon ACP injection, whereas no increase in free carbohydrates was found in the haemolymph. In contrast, ACP had the opposite effect in adult mosquito males: no change in the abdominal glycogen content but an elevation of circulating carbohydrates was observed. There is, thus, a need for further investigation into the ACP signalling system to ascertain the extent of functional overlap with the AKH system.

In the current study, we aimed to address this information gap by examining the AKH and ACP signalling systems of *A. aegypti* which is an infamous disease vector for pathogens, such as yellow fever virus (estimated to cause 200,000 cases of disease and 30,000 deaths each year, with 90% occurring in Africa [22]), Dengue fever, chikungunya, and Zika, as the latest addition to the spectrum of arboviruses, all of which summarily are responsible for a great number of painful infections and death of people following virus transmission from a mosquito bite [23]. *A. aegypti* was selected as a test case since there is already a fair amount of data available from partial investigations into its neuropeptide systems. Furthermore, knowledge of the interaction of bioactive ligands with their cognate receptors is thought to be very helpful for drug research using the GPCRs as a target and aiming for the development of selective bio-rational insecticides [24,25,26].

Historically, the *A. aegypti* peptides were first predicted from genomic work [27], and the AKH and ACP precursors were then cloned, although the ACP was named Aedae-AKH-II at the time [28]. The presence of mature Aedae-AKH-I and corazonin was shown by direct mass profiling, but evidence of the ACP peptide was not found in brain or CC tissue, possibly on account of a low concentration [29]. One Crz receptor (CrzR) and variants of the AKHR and ACP receptor (ACPR) were cloned from *A. aegypti* [28,30,31]. Each receptor was shown to be very selective, accepting only the cognate peptide, thus signifying that there are three separate and independent endocrine systems active in this mosquito species.

Structure-activity relationship (SAR) studies supply information about how a ligand interacts with its cognate receptor, especially which amino acid residue of the ligand is important for this interaction. Thus, by replacing each residue successively with simple amino acids, such as Gly or Ala, or by using bio-analogues (naturally occurring peptides with one or two known differences from the endogenous ligand), the importance of each amino acid’s side chain can be probed with respect to functional or receptor binding outputs. Such SAR studies were conducted on the AKH system in a few insects and a crustacean either by measuring physiological actions in vivo [32,33] or in a mammalian cellular expression system [34,35]. For *A. aegypti* very informative data sets exist on the AKH and the ACP signalling system [36]: for the first time ever, SAR studies were performed with an ACP ligand/receptor system, and it was clear that the chain length of the ligand is important for receptor activation (no activity with 8 but with 10 amino acids), as well as C-terminal amidation, and aromatic amino acids (Phe and Trp at positions 4 and 8, respectively). A 400- to 500-fold loss of activity was measured when the N-terminal pyroglutamate (pGlu) or Thr at position 3 was replaced individually by Ala. Replacements at positions 2, 5, 6, and 7 were well tolerated, so it seems that they are not as intricately involved in receptor activation. For the *A. aegypti* AKHR, it was shown that, in general, the C-terminal portion of the AKH octapeptide, excluding the Trp at position 8, is not as critical for activation as are N-terminal positions 2, 3, and 4; and a longer chain length from ten amino acids appears permissible [36]. Hence, there is some information available on ligand-receptor requirements for two of the three GnRH superfamily signalling systems of *A. aegypti*.

In conjunction with nuclear magnetic resonance (NMR) data on the secondary structure of AKHs [37,38] and knowledge of the receptor sequence, molecular dynamics (MD) methods can derive models of how the ligand interacts with its receptor [39,40,41]. Such data are missing for the yellow fever mosquito and the ACP signalling system of any insect. The current study, hence, investigates the properties of the ligands Aedae-ACP (pEVTFSRDWNA-NH_2_) and Aedae-AKH (pELTFTPSW-NH_2_) by NMR spectroscopy in SDS (sodium dodecyl-d_25_ sulfate) micelle solution to determine their secondary structures, models the interaction of the ligands with the respective receptor and evaluates the predicted models by using the published structure-activity data. Note that Aedae-ACP is the same peptide sequence as in *A. gambiae* [1], and in many insect species [5]. Aedae-AKH is not only endogenous in *A. aegypti* and in the alderfly, *Sialis lutaria* [42], but is also encoded in the genome of *L. migratoria* [43] and *Schistocerca gregaria* [44]. In the context of examining the AKH signalling system of the desert locust, the solution structure of Aedae-AKH was previously determined along with the other two endogenous AKHs of *S. gregaria* [45]. This supplies us with valuable comparative information.

## 2. Materials and Methods

### 2.1. NMR Spectroscopy

NMR samples were prepared by dissolving the dry peptides in 150 mM sodium dodecyl-d_25_ sulfate (SDS), 20 mM phosphate buffer pH 4.5, 0.05 mM TSP (trimethylsilylpropanoic acid), and 10% *v*/*v* D_2_O, to reach a final peptide concentration of 2 mM. All NMR spectra were obtained at 310 K on an 800 MHz Bruker Advance II equipped with a 5 mm TCI cryoprobe. ^1^H homonuclear TOtal Correlation SpectroscopY (TOCSY) (mixing time τ_mix_ = 80 ms) and Nuclear Overhauser Effect SpectroscopY (NOESY) (τ_mix_ = 150 and 300 ms) were acquired with 512 points and 1 k points in F1 dimension, respectively, and 4 k points in F2 dimension, between 16 and 32 transients were accumulated with 1.5 s recycle delay and multiplied with squared sine bell functions shifted by 90°. The ^1^H spectral window was set to 9600 Hz. ^13^C-1H, Heteronuclear Single Quantum Coherence (HSQC) experiments were performed with 256 points in F1 dimension and 4 k points in F2 dimension, and 64 transients were accumulated with 2 s recycle delay. The ^13^C spectral window was set to 33,200 Hz. Non-uniform sampling ^15^N-^1^H HSQC experiments were performed with 25% of 128 points in F1 dimension and 4 k points in F2 dimension, 1024 transients were accumulated with 1.5 s recycle delay. The ^15^N spectral window was set to 3240 Hz.

All data dimensions were zero-filled to twice the respective Free Induction Decay (FID) size. ^1^H chemical shifts were referenced to TSP at 0 ppm, and ^13^C and ^15^N were indirectly referenced to the ^1^H reference frequency. Data were processed in TopSpin (Bruker) and analysed using the CCPNmr analysis program [46]. Backbone and side chains were assigned using all experiments.

### 2.2. Structure Calculations

The NOESY cross-peak assignments were used to generate distance restraints for the structure determination. These distance restraints were supplemented with dihedral angle restraints predicted with DANGLE [47] from Hα chemical shifts. A standard CNS 1.1-based protocol was employed using the ARIA 2.2 interface [48]. The 10 lowest energy structures were refined in a water shell and evaluated with MolProbity [49].

### 2.3. Molecular Dynamics of Ligand

The output from the NMR structural calculations was used as input to GROMACS version 2018.6 [50] for extended MD simulation in water and dodecyl phosphocholine (DPC) micelle. For the water simulations, a box containing the peptide, chloride to neutralise any charge and 7000 water molecules was constructed. The single-point charge water model was used. For the membrane simulations, the lowest energy structure from the simulations in water was placed in the centre of a 7 nm cubic box filled with approximately 10,000 water molecules and a micelle of 50 DPC molecules [51]. The micelle was translated so that, using periodic boundary conditions, half the micelle was at the bottom of the box and the other half was at the top. Energy minimisation was carried out using the steepest descent method for 10,000 steps to a tolerance of 10 kJ mol^−1^. A series of constant pressure, temperature, and number of particles (NPT) equilibration steps were performed to solvate the peptide before the final MD simulation for 50 ns under constant temperature, volume and number of particles (NVT) conditions at 300 K.

The OPLS-AA/L force field [52] was used to describe the molecule bond energies. All bonds were constrained using the LINCS algorithm [53]. A cut-off of 1.0 nm was used for van der Waals and electrostatic interactions. Following equilibration, MD was performed for 50 ns at 300 K under NVT conditions. For each simulation, 100 snapshots were collected over the course of the simulation. Cluster analysis of the resulting structures was performed using the linkage algorithm of GROMACS with a cut-off of 0.1 nm on the backbone atoms.

### 2.4. Construction of Receptor Models, Ligand Docking and Molecular Dynamics

The primary sequence of the adipokinetic hormone/corazonin-related peptide receptor, *Aedes aegypti* ACPR-I (Genbank MF461644; protein: AVA08868.1), was used to construct the 3D structure of the receptor at the atomic level. The primary sequence of the *Aedes aegypti* AKH receptor was taken from Genbank MF988326 (protein AV109459.1). Swiss-Model [54] was used to search for target templates. For both receptors, the NMR structure of active β_2_ androgenic receptor, 6kr8.1.A, was selected. For the inactive or ‘open’ model of Aedae-ACPR the 5D5A X-ray crystal structure of the β_2_ androgenic receptor at 100 K was chosen. For Aedae-AKHR the Xray structure 6tpk was selected as a template of the inactive conformation. The same website was then used to construct 3D models of the receptors, based on these two templates. The resulting structures were imported into Maestro 13.1 [55] for visualisation, ligand docking, and MD calculations. The proteins were first prepared using the Maestro Protein Preparation Wizard, and the quality of the models was checked with Ramachandran plots. The Maestro Glide module was used with SP-Peptide precision to find the best poses of the ligand in the receptor binding pocket. The pose with the best glide score was used for MD simulation. For this the receptor/ligand construct was imbedded in a (1-palmitoyl-2-eleoyl-3-phosphocholin) (POPC) membrane, neutralised with Cl^−^ and solvated with SPC water, using the membrane setup of Desmond [56]. Molecular dynamics was performed using Desmond and the free energy of binding was calculated using MM/GBSA (Prime version 2019, Schrödinger, LLC, New York, NY, USA, 2019).

## 3. Results and Discussion

### 3.1. Aedae-ACP NMR Results

For small and partially folded peptides with affinity for lipid membranes, NMR is a very well-suited structural technique compared to cryo-EM or X-ray techniques which rely on ordered or large homogenous copies of folded molecules. The NMR assignments and chemical shift of Aedae-ACP in SDS micelle solution are given in Table 1. Some ideas of the secondary structure and flexibility of the peptide can be obtained by comparing these chemical shifts to those of the same residue in a random coil environment [57,58]. These chemical shift indices are plotted in Figure 1a. Both the H_α_ and H^N^ protons are consistently shifted up-field suggesting some type of turn structure [59]. This is consistent with the review of Tyndall et al. [60] of over 100 mammalian GPCR ligands, which all had a turn structure. The chemical shift indices are also consistent with our results for members of the AKH family, such as Melme-CC [61], Declu-CC [61], Dappu-RPCH [40], Schgr-AKH-II [40], Anoga-HrTH [39] and including Aedae-AKH [39].

The NMR chemical shifts can also be used to estimate the flexibility of the peptide [57]. The results (Figure 1b) show that the peptide is quite flexible with an order parameter (*S^2^*) ranging from 0 to only 0.36. A perfectly ordered (rigid) structure has an *S*^2^ of 1. The Aedae-ACP order parameter results are the same as those found for Aedae-AKH (*S*^2^ = 0.1–0.3) [45], Schgr-AKH-II (*S*^2^ = 0.1–0.4) [45] which are similar to that of the crustacean member of the AKH family, Dappu-RPCH (*S*^2^ = 0.1–0.25) [40]. However, the order parameter contrasts with the rigid conformations of a number of insect AKHs; Melme-CC (*S*^2^ = 0.85) [61], Declu-CC (*S*^2^ = 0.7–0.9) [61], Locmi-AKH-I (*S*^2^ = 0.9) [45] and Anoga-HrTH (*S*^2^ = 0.7–0.8) [39]. As expected, Aedae-ACP is more ordered in the middle of the peptide than at the termini. The NMR-derived root mean square fluctuation (RMSF) of the Aedae-ACP backbone (Figure 1c) also shows that the peptide is flexible, with RMSF values ranging from 3.5 to 8.7 Å. The RMSF values of the more rigid Melme-CC [61], Declu-CC [61] and Anoga-HrTH [39] range from 0.3 to 1.7 Å. In contrast, Dappu-RPCH has RMSF values ranging from 5 to 8 Å [40].

The NMR restraints were used to search the conformational space of Aedae-ACP. The resulting structures were grouped according to the similarity of their backbone conformation. The root conformer of the largest cluster was placed in a solution with a DPC micelle and MD was performed for 50 ns. Figure 2a shows an overlay of the largest cluster. This cluster is extended at the C-terminus but has a β-turn around threonine at position 3 from the N-terminus. This conformation had no restraint violations and agreed with the NMR chemical shift results.

### 3.2. Structural Comparison of Aedae-ACP and Aedae-AKH

Previously, we had determined the solution structure of Aedae-AKH as one of the locust AKHs, using NMR-restrained molecular dynamics [45]. Here we compare the predominant solution conformations of Aedae-ACP and Aedae-AKH (Figure 2b). Aedae-AKH has a more linear structure than Aedae-ACP, which has a pronounced β-turn at Thr^3^. This β-turn brings Phe^4^ and Trp^8^ onto the same side of the peptide. In contrast, Aedae-AKH has a proline at position 6, which introduces a turn, and again, brings Trp^8^ to the same side as Phe^4^. Hence, these two peptides have the same orientation of Phe^4^ and Trp^8^, albeit from different mechanisms. It is well known that these two aromatic residues are the most conserved AKH/ACP ones in AKH and ACP peptides, and are reportedly essential for receptor activation [8,44]. It is noted, however, that these two peptides are very flexible. The *S*^2^ order parameter of both Aedae-ACP and Aedae-AKH is only 0.36.

### 3.3. Aedae-Receptor Homology Modelling

Homology modelling was used to construct two 3D models of each of the Aedae-ACP and Aedae-AKH receptors. Two templates were used for each receptor, one with the receptor in an active state (X-ray structure 6kr8.1.A) and one with the receptor in an inactive state (X-ray structure 5D5A for Aedae-ACPR and 6tpk for Aedae-AKHR). The quality of all 4 models was tested using Ramachandran plots. Each had torsion angles in the disallowed region of conformational space, but these outliers disappeared upon MD optimisation of the models. Both Aedae-ACPR and Aedae-AKHR have the conserved residues, typical of the rhodopsin superfamily of GPCRs, namely (using the Ballesteros and Weinstein numbering system [62]): N^1.50^; D^2.50^, P^2.59^; C^3.25^, the DR^3.50^x motif (DRY for the AKHR and DRC for ACPR), W^4.50^; P^5.50^; F^6.44^, the CWxPY motif (CWxP^6.50^Y), and the NPxxY^7.53^ motif. These conserved residues are essential for maintaining the structure of the receptors and for their activation. Figure 3 shows the optimised structures of the two receptor models [63].

Blind docking was performed on each of the ligands and receptor models of Aedae-ACPR. In each case the same binding pocket was found, which corresponded to the binding region found for the Dappu-RPCH [40], Melme-CC [61], Declu-CC [61], Schgr-AKH-II [45] and Anoga-HrTH [39] receptors. Glide scores ranged from −6.6–−16.9. The docked structures with the highest glide scores were placed in a POPC membrane and MD performed for 200–2200 ns.

### 3.4. Aedae-ACPR Models

The two homology models of Aedae-ACPR are given in Figure 3a,b. Essentially, their backbone structures are the same except for small, but essential, relative movement of the helices. Also, the orientation of ECL2 and ICL3 are different. In the active state, ECL2 lies over the binding pocket, preventing the egress of the ligand. At the same time, ICL3 moves away from the G-protein binding site. In the inactive state, ECL2 moves to the side allowing the ligand access to the binding site, and ICL3 projects further from the membrane.

A more noticeable difference in the two models is shown by the ionic lock between Arg^205^ and Glu^299^ (Figure 3c). In the inactive model, the lock is closed with an inter-residue distance of 5.56 Å, while in the active model, the lock is open with an inter-residue distance of 13.5 Å. Aedae-ACPR also has a number of other switches found in class A GPCRs. These include the DRC switch, the tyrosine toggle switch, and the hydrophobic connector shown in Figure 4.

Water is integrally involved in the action of GPCRs. The crystal structures of B2AR all have internal water molecules, and it has been postulated that these waters are necessary for ligand and G-protein binding [64]. A plot of water density (above bulk) in the active model of Aedae-ACPR is shown in Figure 4d. In the figure, water is clustered, internally, at both ends of the receptor where the ligand and G-protein bind.

#### 3.4.1. Aedae-ACP Docked to Aedae-ACPR

With the generated Aedae-ACPR models, ligand docking was performed to study specific interactions, for the first time, between an ACPR and its peptide ligand, Aedae-ACP. Figure 5a shows the protein root mean square deviation (RMSD) of the 6kr8 (active) model of Aedae-ACPR with Aedae-ACP docked. The blue curve is the displacement of the receptor C_α_ atoms relative to the starting structure and the red curve is the ligand heavy atom displacement relative to the receptor. The ligand-heavy atoms move ~3 Å from their original position to their final position. This happens in the first 0.5 ns of the simulation. The C_α_ atoms of the receptor, being larger, take ~40 ns to adopt their final position. Note the two curves run essentially parallel to each other, indicating that the ligand does not move relative to the receptor. The model is well equilibrated as the RMSD does not change over time.

The RMSF of the C_α_ atoms of the active receptor model is shown in Figure 6a. Apart from ICL3, which fluctuates 4.8 Å, there is very little change in the conformation of the receptors. The conformation of the trans-membrane helices remains fairly rigid. Protein residues that interact with the ligand are shown as vertical green lines. Aedae-ACP interacts with ECL1 and the adjacent residues of H2 and H3. Another two regions of ligand/protein contact are ECL2, H5, H6, and ECL3. The RMSF profile of Aedae-ACPR is similar to those found for several AKH insect receptor models [40].

The RMSD profile of the 5D5A (inactive model) model (Figure 5b) is very different. Here, the ligand and the receptor continue to move throughout the 2.2 μs simulation. Essentially the inactive conformer is slowly changing into the active conformer. This is shown by the change in RMSD between the inactive and active models over time. Note, however, that the protein continues to move but the ligand settles down, indicating that it is in the correct binding orientation at the start of the simulation.

The RMSF of the C_α_ atoms of the 5D5A receptor model (Figure 6b) shows a similar pattern to the 6kr8 model, albeit with larger fluctuations of ICL3. The ligand interacts with the same region of the protein as the 6kr8 model.

The RMSF of the ligand (Appendix A) shows how the ligand atoms interact with the protein and their entropic role in the binding event. Here the protein-ligand complex (Aedae-ACPR–Aedae-ACP) was first aligned onto the protein backbone and then the RMSF of the ligand atoms was calculated, so that the RMSF indicates the movement of the ligand within the receptor binding pocket. The RMSF of both models is very similar. There is very little fluctuation (~1 Å) of the ligand backbone and sidechains, except for Phe^4^ and Arg^6^. The ACP N-terminus of the active model fluctuates more than that of the inactive model. This is because the ACP C-terminus projects into the core of the receptor, while the N-terminus is towards the extra-cellular region of the receptor. The C-terminus of both models has ~2.5 Å fluctuations. Of interest is the lack of motion of Trp^8^, which indicates that it is tightly bound in the active site. These results indicate that the ligand does not move substantially in the binding pocket.

Appendix A gives more detail about which ACP receptor residues interact with Aedae-ACP. Here protein contacts with the ligand are normalised over the trajectory such that a value of 1.0 means that the contact is maintained for 100% of the simulation. Since a particular residue may have more than one contact, values above 1 are possible. The interactions are also categorised by type: H-bond, hydrophobic, ionic, and water bridges. In the inactive state, Aedae-ACP has multiple, short-term, interactions with its receptor. However, only a few are persistent: Asn^274^ has a persistent H-bond and water bridge with the ligand; and Arg^187^ forms a H-bond for 30% of the simulation, a hydrophobic interaction for 50% of the simulation, and a water bridge for the entire simulation. Also, there are H-bonds and water bridges with the receptor: Pro^266^ and Pro^267^ H-bond to the ligand for the entire simulation; Val^246^ has a water bridge to the ligand.

In the active state, Aedae-ACP has 14 persistent contacts with the receptor (Appendix A). Asn^274^, Arg^187^, and Val^246^ contacts are still present, but there are now persistent contacts with Asp^153^; Ala^188^, Leu^192^ and Ser^195^ on H3; Asn^236^; Gln^247^, Phe^261^, Met^269^, and Cys^277^ on ECL2; and Trp^350^, Tyr^357^, Thr^357^ and Tyr^360^ on H6. Since this is the first modelling and in silico docking of ACPR and ACP as ligands, we have no other comparisons for discussion. However, a study on a GPCR from the stick insect *Carausius morosus* (touted as an AKHR but without empirical evidence) also noted that residue Arg^187(3.32)^ is critical for AKH binding to the putative AKH receptor [65]. In the ligand docking of desert locust AKHs to the cognate AKHR model, Arg of Schgr-AKHR was also found to be instrumental/involved in the binding of Schgr-AKH-II [45].

The simulation interaction diagram (SID) gives details of which ligand atoms interact with the receptor. In Figure 7 the interactions between Aedae-ACP and Aedae-ACPR are colour-coded such that H-bonding is depicted in purple, while π-cation interactions are shown in red. For the duration of the simulation, contacts between the ligand and receptor are formed and broken. This is recorded as a % of the simulation time. Only interactions that persist for more than 30% of the simulation are shown. The surface of the receptor is shown as a solid line, again colour coded, polar, hydrophobic, etc., according to the nature of the receptor residues. The active model (Figure 7a) has many interactions with the central residues of the ligand. The serine OH, H-bonds to Gln^247^, Met^269^, and Val^246^ for the entire simulation. The SID of the active model has a number of water molecules in the binding site, which Yuan et al. [64] have postulated are essential for receptor activation. These water molecules bridge between the ligand and the receptor. Interestingly, Trp^8^ of Aedae-ACP, which is postulated to be essential for binding, interacts with Arg^187^ in both ACP receptor models. The inactive model (Figure 7b) does not have as many interactions between the ligand and receptor but does have several internal H-bonds, which help to maintain the conformation of the ligand.

The strength of the ligand-receptor binding is best measured by the free energy of binding (ΔG_bind_) averaged over the course of the trajectory. The free energy of binding of Aedae-ACP bound to the active model of Aedae-ACPR is −137 ± 10 kcal mol^−1^, but only −88 ± 9 kcal mol^−1^ for the inactive model of Aedae-ACPR. It is the closing of the receptor around the ligand that is responsible for the increased binding energy.

#### 3.4.2. Residue Scanning of Aedae-ACPR

The best way to validate a computational receptor model with simulated ligand docking is to follow up with experiments involving the physical GPCR, its cognate ligand, and a variety of other ligands that may be more- or less-suited for interacting with and activating the GPCR according to the modelled data. In the case of Aedae-ACPR and Aedae-ACP, receptor functional activation studies were completed 5 years ago in vivo in a mammalian cell line and the EC_50_ (the concentration of the ligand that produces 50% of the maximum response) was recorded in a typical SAR study via the well-known bioluminescence reporter assay [36]. In the current study, hence, we tried to replicate those experimental results computationally by using our generated Aedae-ACPR models and simulating ligand docking with the same series of ACP analogues used by Wahedi et al. [36] in their heterologous expression of Aedae-ACPR. Computationally, the ligand-receptor binding strength was measured by the free energy of binding (ΔG_bind_) and so an inverse relationship between ΔG_bind_ and EC_50_ was expected between these two parameters.

Figure 8 shows the computational results of sequentially replacing the residues of Aedae-ACP with an alanine: substitution of Val^2^, Thr^3^, Phe^4^, Ser^5^, Arg^6^, and Asn^9^ all decreased the affinity slightly relative to the native peptide. On the other hand, mutation of Asp^7^ increased the binding affinity, while mutation of Trp^8^ decreased the binding affinity by 19 kcal mol^−1^. Also shown in Figure 8 are the EC_50_ values taken from Washedi et al. [36] as a comparison to our computationally derived results—note that EC_50_ values could not be calculated in the case of ACP analogues with Ala^4^ and Ala^8^ substituted for Phe^4^ and Trp^8^, respectively, for those substitutions resulted in very little to no detectable activation of Aedae-ACPR even at a peptide dose of 10 μM. An inverse relationship between ΔG_bind_ and EC_50_ is evident for the Ala-substituted ACP analogues, except for the Ala^4^ and Ala^7^ mutations (Figure 8). This indicates that the proposed model may be virtually correct with a few possible complications. Further modelled results show a good correlation: Aedae-ACP (native ligand) and the Ala^9^ substituted ACP has the lowest EC_50_ values and the highest binding affinities, while the replacement of Trp^8^ with Ala, had the lowest binding affinity thus, fitting with the receptor assay data of an inactive peptide even at a very high dose. This begged the question as to why the modelled/calculated binding affinity failed to predict the experimental loss of activity upon replacement of Phe^4^ by alanine. In fact, this is not the first occurrence, for the same mismatch was previously established in the simulated residue scanning of the octapeptide Dappu-RPCH with Ala^4^ substitution for Phe^4^ and activation of Dappu-RPCHR [40]. In the same vein, the Ala^7^ mutation of Aedae-ACP had the highest binding affinity in the current study (Figure 8), whereas this analogue had only the same level of receptor activation in receptor assays as the Ala^2^, Ala^5^, and Ala^6^ mutants [36] (Wahedi et al. 2019). Evidently, ligand binding is only one of the steps that may lead to receptor activation or not. This notion of a ligand binding without activating the receptor and the subsequent signal transduction cascade is borne out by pharmacological studies with inhibitors (antagonists) and when simply considering the idea of drug “efficacy”, loosely defined as the size or strength of a response produced by a particular agonist in a particular tissue [66]. Thus, just because a ligand has affinity it does not necessarily mean that it will have efficacy; for example, a simple antagonist will have affinity but an efficacy of zero. When it comes to drug studies, the ability to bind to a receptor may determine the ability to produce a response and to some extent the size of that response; however, the two are seldom linked in a linear fashion [66]. We, therefore, conclude that certain changes to the peptide ligand may result in binding with the ACPR but not in activating the receptor and that our proposed in silico models are not fully able to distinguish between receptor binding and ligand efficacy. It is advisable, therefore, to use complementary methodologies to rigorously test model predictions. A point in the case can be found in Jackson et al. [67] where a computationally predicted antagonist of a locust AKHR was, indeed, proven to be a competitive inhibitor of AKH through the use of in vivo biological assays.

Analysis of the current data shows that, although Trp^8^ is 79% buried within the receptor, mutation to Ala^8^ leads to an increase in solvent-accessible surface area (Δ_SASA_ = 90 Å^2^). It is also interesting to note that Trp^8^ has a surface complementarity of 0.88 where a complementarity of 1 means the two surfaces match perfectly and 0 means the two surfaces have no complementarity.

Wahedi et al. [36] further tested the receptor activation ability of a number of other synthetic and natural analogues of ACP, and those same analogues were modelled and docked to the Aedae-ACPR models in the current study. C-terminal amidation was found to be critical for receptor activation. MD of the non-amidated peptide with the inactive receptor model had free energy of binding some 50 kcal mol^−1^ greater than the amidated peptide. This is mainly due to the very strong H-bonding between the free acid and Asp^153^. On the other hand, binding to the active model was ~20 kcal mol^−1^ less stable than native ACP. This is the same binding as the Trp^8^ mutation. In the bioassay, both Trp^8^ and C-terminal amidation were essential for receptor activation.

MD of pETFSRDWNA-NH_2_, an internally truncated ACP by deletion of valine in the second position, had free energy of binding ~30 kcal mol^−1^ less than native ACP and, so by analogy with [Ala^8^]ACP and ACP[COOH], is predicted to be inactive. These results of the current study, indeed, corroborate the receptor assay findings of [36].

### 3.5. Aedae-AKHR Models

The AKHR of *A. aegypti* has never before been modelled. Here, we attempted to remedy this and to use the information to determine whether there could be cross-activity between AKHs and ACPs. The overlay of the two models (active and inactive) of Aedae-AKHR is shown in Figure 9. The two structures are very similar but the transmembrane helices of the 6kr8 (active) model are twisted relative to the 6tpk (inactive) model. This opens the intracellular side of the active receptor model and closes the extra-cellular side. At the same time the extra-cellular loops move, as shown in Figure 9b. ECL1 does not move. In the inactive model, ECL2 and ECL3 are positioned away from the central axis of the receptor, allowing free access of the ligand to the binding site. In the active model, ECL2 and ECL3 have moved over the top of the receptor preventing the ligand from leaving. This is particularly so for ECL2. At the same time, the orientation of the helices and loops on the intracellular side of the receptor are different (Figure 9c). In the inactive model the helices and loops close the binding site of the G-protein, while in the active model, ICL1 and ICL3 have opened up allowing the G-protein free access. This movement of the helices and loops is typical of GPCR activation [63].

Like Aedae-ACPR, Aedae-AKHR has a number of switches. There is the conserved DRY motif on TM3 but there is no conserved Glu^6.30^ on TM6. Instead, there is a lysine which can H-bond to Arg^3.50^.

Appendix A shows the protein RMSD of the 6kr8 (active) model of Aedae-AKHR with Aedae-AKH docked. The blue curve is the displacement of the receptor C_α_ atoms relative to the starting structure and the red curve is the ligand heavy atom displacement relative to the receptor. The ligand-heavy atoms move ~2 Å from their original position to their final position. This happens in the first 0.5 ns of the simulation. The receptor C_α_ atoms have not settled down even after 300 ns.

The RMSF of the C_α_ atoms of the active receptor model is shown in Appendix A. The conformation of the trans-membrane helices remains fairly rigid. The loop regions fluctuate ~2 Å, except for ICL3, which fluctuates ~5.2 Å. Protein residues that interact with the ligand are shown as vertical green lines. The ligand interacts with ECL1 and the adjacent residues of H2 and H3. The ligand also contacts receptor residues in ECL2, H5 and H6, and ECL3. The RMSF profile of Aedae-AKHR is like that of Aedae-ACPR and all other AKH receptors we have studied [37,38,45,61].

The RMSD profile of the 6tpk (inactive model) model (Appendix A) is different. Here, both the ligand and the receptor settle down very rapidly and then do not change position for the rest of the simulation. This is different from the Aedae-ACPR simulation, where the inactive model continued to change conformation for 2.2 μs. The receptor and ligand curves run essentially parallel to each other, indicating that the ligand is not moving relative to the receptor. The model is well equilibrated as the RMSD does not change over time.

The only notable difference between the C_α_ atom RMSF profile of the 6tpk (inactive) and 6kr8 (active) receptor models of Aedae-AKHR, (Appendix A) is the large fluctuations of ICL2. This is the loop that moves to trap the ligand in the receptor binding site. The ligand interacts with the same region of the protein as the 6kr8 model. In both models, ICL3 moves substantially to allow the G-protein access to the receptor.

Aedae-AKH does not move substantially in the binding pocket of Aedae-AKHR, as shown by the ligand RMSF (Appendix A). In the 6kr8 model of Aedae-AKHR, the backbone atoms of docked Aedae-AKH, do not fluctuate significantly. There are ~2 Å fluctuations of all the side chain atoms, except for Trp^8^ and possibly Phe^4^. The lack of motion of Trp^8^ indicates that it is tightly bound in the active site. The ligand atom fluctuations in the 6tpk model are greater than in the 6kr8 model. The largest variation was seen in the Phe^4^ side chain atoms. There is also substantial variation in the N-terminal atoms.

Appendix A shows which receptor residues interact with the ligand, while Appendix A shows which ligand residues interact with the receptor. Combining these two diagrams, the details of the receptor-ligand interactions become apparent. In the two models of the Aedae-AKH receptor, the ligand, Aedae-AKH, binds to the same region of the receptor, but the details of the interaction are different. In the active model, Trp^133^ on helix 2 H-bonds to Pro^6^(CO) for the entire simulation. In the inactive model, Trp^133^ still H-bonds but now to Trp^8^, with which it also has a π-π interaction. In both models, Asn^238^ on H5 interacts with Val^2^ of the ligand. These are the only two common receptor residues, in the two models, that bind to the ligand. All the other receptor-binding site residues are displaced slightly in the two models. Thus, in the active model, Asn^226^ on ECL2, H-bonds to Thr^5^(CO), while in the inactive model, Thr^224^ on ECL2, H-bonds to Thr^3^(CO). Ser^313^ on H6 of the active model has a water bridge with pE(CO) of the ligand, while in the inactive model, Trp^317^ H-bonds to Val^2^(CO) and has a π-π interaction with Phe^4^ and Tyr^316^, and H-bonds to Phe^4^(NH) and Ser^5^(CO). In addition, in the active model, Tyr^233^ on ECL2 has multiple interactions with ligand Phe^4^; Lys^332^ on H7, H-bonds to Thr^5^(OH) and has a water bridge with Phe^4^(CO); and Arg^153^ on H3 H-bonds to Thr^3^(OH). The similarity of binding of the active and inactive Aedae-AKHR models is reflected in their free energy of binding. For the active model, ΔG_bind_ is −137 ± 10 kcal/mol^−1^, while for the inactive model, it is −121 ± 8 kcal/mol^−1^.

In the Ballesteros and Weinstein numbering system, Arg^153^ is denoted 3.32 and corresponds to Arg^269^ of the stick insect (*Carausius morosus*) AKH receptor and Arg^107^ of Schgr-AKHR. In the stick insect, Arg^269^ was found to be critical for ligand binding [65], and in Schgr-AKHR, Arg^107^, H-bonds to Schgr-AKH-II [45]. In addition, Iyison et al. [65] found very stable interactions between the ligand and Gln^7.35^, and Tyr^6.51^ and π-π stacking to Phe^3.36^ and Trp^6.59^ of the stick insect putative AKHR. We do not find the interaction between residue 3.36 and the ligand but there is a hydrophobic interaction between residue 6.59 (Trp^317^) on Aedae-AKHR and Phe^4^ of Aedae-AKH and residue 7.35 (Gln^331^) and Thr^3^(OH). There is also H-bonding and a water bridge between residue Lys^332(7.36)^ and Thr^5^(OH) of Aedae-AKH. In the inactive model of Aedae-AKHR, Glu^130(2.61)^ binds to Thr^5^(CO). In the stick insect residue, 2.61 corresponds to Glu^246^, which binds to Carmo-HrTH-I.

#### 3.5.1. Comparison of Aedae-AKH Bound to Aedae-AKHR and Schgr-AKHR

In addition to being found in the yellow fever mosquito, *Aedes aegypti*, Aedae-AKH is also found in the desert locust, *Schistocerca gregaria*. The conformation of Aedae-AKH is similar when bound to the two receptors Aedae-AKHR and Schgr-AKHR (Figure 10a,b). The binding pocket of Schgr-AKHR consists of a cleft running across the top of the extracellular domain of the receptor between helices 2, 6, and 7 and extracellular loops 2 and 4. Aedae-AKH fits into this cleft with the central portion of the peptide fitting into the binding pocket and the two termini pointing outwards (Figure 10d). The ΔG_bind_ of Aedae-AKH to Schgr-AKHR is −88 kcal/mol^−1^. This contrasts with Aedae-AKH binding to Aedae-AKHR, where the N-terminus of the ligand extends into the receptor with Trp^8^ at the surface of the receptor (Figure 10c). This different orientation of the ligand in the binding pocket of Schgr-AKHR and Aedae-AKHR is the reason for the much higher free energy of binding of Aedae-AKH to Aedae-AKHR. It may also account for the promiscuity of Schgr-AKHR [45].

#### 3.5.2. Binding of Natural Arthropod AKH Analogues to Aedae-AKHR

To test and validate our Aedae-AKHR receptor models, we modelled the docking of various AKH analogues to the receptor, including the native octapeptide Aedae-AKH (pELTFTPSWamide). Once again, we modelled insect bioanalogues (i.e., naturally occurring AKH ligands), as well as specifically modified ligands, that were previously tested in an in vitro functional Aedae-AKH receptor activation assay by Wahedi et al. [36]. In this way (as for our ACPR models above), we could directly compare and assess the accuracy of the computationally generated ligand-receptor (Aedae-AKHR-1A) models. Several authors have found an aromatic residue at position 8 to be essential for AKH receptor activation and the biological activity of AKH peptides. We have simulated this by replacing Trp^8^ with alanine. The results show a much-reduced binding of the mutated analogue, with a Δ_binding_ of −89 kcal mol^−1^, compared to −120 kcal mol^−1^ for the native peptide. The main reason for this difference in binding energy is the lipophilic interaction of Trp^8^. In the native Aedae-AKH peptide, Trp^8^ sits in a lipophilic pocket of the Aedae-AKHR created by Trp^139^, Trp^133^, Phe^208^, and Leu^129^. The Δ_lipo_ for this is −30 kcal mol^−1^. For the Ala^8^ mutation, Δ_lipo_ is −23 kcal mol^−1^ and the ligand has a completely different orientation in the binding pocket (Appendix A). There is no interaction with Arg^153^. Thus [Ala^8^]AKH does not interact with the receptor residues necessary for activation. This mutation was not tested by Wahedi et al. [36], but a designed, synthetic analogue of Lacol-AKH, Lacol-AKH-7mer (pELTFTSS-NH_2_), was tested in the receptor bioassay and found to be only 7% active at the highest concentration of 10 µM tested. Thus, this peptide did not activate Aedae-AKHR. Since Lacol-AKH-7mer differs from [Ala^8^]Aedae-AKH in being C-terminally truncated (no Trp^8^) and having Pro^6^ replaced by Ser^6^, we also simulated the binding of this 7-mer peptide to Aedae-AKHR. Here a Δ_binding_ of only −78 kcal mol^−1^ was obtained, a difference of some 43 kcal mol^−1^ less compared to native Aedae-AKH. Interestingly, the 7-mer form of Lacol-AKH had a better solvation energy (16 kcal mol^−1^) than Aedae-AKH but a poorer Coulombic (−27 kcal mol^−1^) and van der Waals (−24 kcal mol^−1^) interaction. Lacol-AKH also had a weak interaction with Arg^153^.

To better understand the importance of N-terminal residues, Wahedi et al. [36] measured the activity of Erysi-AKH, a peptide that occurs naturally in the dragonfly *Erythemis simplicicollis*. This peptide differs from Aedae-AKH at position 3 (threonine to asparagine exchange). This substitution resulted in a 310-fold reduction in the receptor activity study. The Δ_binding_ of Erysi-AKH was calculated in the current study to be 38 kcal mol^−1^ less than Aedae-AKH. Further substitution at position 2 (leucine to valine) as the peptide code-named Libau-AKH, a natural peptide isolated from the dragonfly *Libellula auripennis*, resulted in an activity of 1.7-fold less than Erysi-AKH. Docking of Libau-AKH to Aedae-AKHR had free energy of binding 1.7 kcal mol^−1^ less than Erysi-AKH. An AKH from the firebug, *Pyrrhocoris apterus* (Pyrap-AKH), also has two amino acid substitutions relative to Aedae-AKH (N^3^ and N^7^ vs. T^3^ and S^7^) and relative to Libau-AKH (L^2^ and N^7^ vs. V^2^ and S^7^). In the Aedae-AKHR activation bioassay, Pyrap-AKH was found to be more active than Libau-AKH but less active than Aedae-AKH and Erysi-AKH. Our simulation results are at odds with this observation. The Δ_binding_ of Pyrap-AKH was −153 kcal mol^−1^, which is even more stable than Aedae-AKH. In all, Pyrap-AKH has 9, 11, and 12 kcal mol^−1^ more favourable Coulombic, lipophilic, and van der Waals energies with the Aedae-AKH receptor than Aedae-AKH. Hence, based on the free energy of binding alone, Pyrap-AKH should be more active than Aedae-AKH. Looking at the details of Pyrap-AKH binding, however, it is oriented differently in the binding pocket to Aedae-AKH. It does not interact with the same receptor residues identified previously as essential for receptor activation, which is why it is not active in the receptor assay. On the other hand, there is a very stable interaction between the mutated asparagine in position 7 and the receptor residues Arg^153^, Tyr^309^, Tyr^310^, Gln^204^, and Tyr^237^. It is this strong H-bonding between Asn^7^ on the ligand and the five receptor residues that accounts for the large free energy of binding. This dichotomy again illustrates the disconnect between binding affinity and activity as discussed above.

The importance of proline in position 6 was tested using Hipes-AKH-I from the sphingid moth, *Hippotion eson*. This peptide has the same sequence as Aedae-AKH, except that the proline in position 6 is changed to serine. In the receptor bioassay, Hipes-AKH-I had a very similar activity to Aedae-AKH and hence Wahedi et al. [36] concluded that position 6 is not very important for receptor activation. In our simulation, Hipes-AKH had free energy of binding of −85 kcal mol^−1^, which is better than Lacol-AKH and Erysi-AKH but is less than Aedae-AKH; this result concurs with that of Wahedi et al. [36]. Hipes-AKH does bind to Arg^153^, previously identified as essential for activity.

Finally, Wahedi et al. [36], used the AKH analogue, Tabat-AKH, from the black horsefly, *Tabanus atratus*, where glycine is substituted for serine at position seven, to test the importance of this position. Here they found that Tabat-AKH was 30% more active than Aedae-AKH. Our simulation results, however, show that, while Tabat-AKH has very similar interactions with the receptor as Aedae-AKH, its free energy of binding to the receptor is only 0.5 kcal mol^−1^ better. Given the standard deviation in the free energies of binding, this result is in accord with the bioassay results. Also, Tabat-AKH does interact with Arg^153^.

### 3.6. Docking of Aedae-AKH and Aedae-ACP to the Non-Cognate Receptor

*Aedes aegypti* has two well-characterised signalling systems with the ligands AKH and ACP. The ACP system appears to be more prescriptive than the AKH system in that it does not tolerate shorter peptides, whereas the AKH system is activated by a decapeptide [36].

The selectivity of Aedae-ACPR and Aedae-AKHR for their cognate peptides was checked by cross-binding of Aedae-ACP to Aedae-AKHR and Aedae-AKH to Aedae-ACPR. Docking of Aedae-AKH to Aedae-ACPR resulted in multiple poses with glide scores ranging from −8.7 to −2.3. Contrary to all other studies on AKH peptides, here the peptide did not have a β-turn but only a kink at proline. The ligand was not found in the receptor binding site but across the opening with the Trp^8^ residue projecting into the binding site (Appendix A). MD of the docked ligand had a free energy of binding of −129 ± 6 kcal mol^−1^, which is a rather high free energy of binding. The simulation interaction diagram shows that AKH binds strongly to residues Arg^187^, Glu^164^, Tyr^192^, Asn^274^, Arg^245^, Tyr^353^, and Thr^357^. These are all Aedae-ACP receptor residues outside the binding pocket we had established with Aedae-ACP (see above). The N-terminal pyroglutamic acid of Aedae-AKH had water bridges to a number of Aedae-ACP receptor residues (364, 367, and 368) and the tryptophan side chain sits in a positively charged pocket. All these interactions make for strong ligand/receptor binding but in the wrong place for receptor activation.

Aedae-ACP docks to Aedae-AKHR with glide scores ranging from −10 to −4. The best pose had a ΔG_binding_ of −80 kcal mol^−1^, which is low. The ligand is oriented with the C-terminus projected inside the receptor and the N-terminus pointing out of the binding pocket (Appendix A). The ligand interacts with receptor residues Ser^313^, Tyr^316^, Asn^325^, Asn^327^, Gln^328^ and Gln^331^. None of these residues are essential for receptor activation. Hence the simulation results predict that Aedae-ACP should not activate Aedae-AKHR.

Both docking attempts are pointing clearly to two independent signalling systems with no overlap.

## 4. Conclusions

In this study, homology modelling was used to construct 3D models of the AKH and ACP receptors from the yellow fever mosquito, *Aedes aegypti*. At the same time, NMR-constrained MD was used to determine the preferred conformation of Aedae-ACP in a lipid micelle environment. The structure of Aedae-ACP, together with our previously determined structure of Aedae-AKH, were then used to identify the ligand binding pockets of their respective receptors.

Aedae-ACP was found to have a pronounced β-turn at Thr^3^, which brought Phe^4^ and Trp^8^ onto the same side of the peptide. This conformation was stabilised by intramolecular hydrogen bonds. Aedae-AKH had a more linear structure, but a proline kink at position 6 also moved Trp^8^ to the same side as Phe^4^. Phe^4^ and Trp^8^ are reportedly essential for receptor activation, so it is interesting that these two residues had the same orientation, albeit arising from different mechanisms, in both ligands.

The structure of both receptors was typical of class A, GPCRs, with 7 transmembrane helices and an eighth helix lying in the plane of the membrane’s inner surface. Each had a disulfide bridge between helix 3 and ECL2 and the conserved residues typical of this class of receptor. The requisite switches and locks were found except that Aedae-ACPR had a DR^3.50^C motif instead of the normal DRY motif.

The binding pocket of Aedae-ACPR comprised residues from all the helices, except helix one, and ECL2. Ser^194(3.39)^ defined the bottom of the binding pocket with ECL2 closing the top. In the inactive model, ECL2 lay to the side allowing the ligand access to the binding pocket, and then moved over the ligand upon receptor activation, preventing egress of the ligand.

Ligand residue substitution studies show that strong ligand-receptor binding is a necessary but not sufficient condition for receptor activation. It appears that binding to Arg^153^ is also essential for receptor activation of both Aedae-ACPR and Aedae-AKHR. This is the same residue that was postulated to be essential for the activation of the stick insect (*C. morosus*) and the desert locust (*S. gregaria*) AKHRs. Docking studies also showed why cross-activation of Aedae-ACPR and Aedae-AKHR by Aedae-ACP and Aedae-AKH is not possible.

The results of this NMR-MD study are the first of its kind to give more insight into the adipokinetic hormone/corazonin-related peptide signalling systems of the yellow fever mosquito, *Aedes aegypti*, and is novel in providing a plausible comparison between the two similar receptors (ACPR and AKHR) and the reason why they bind and are activated by their respective cognate ligand and are not activated by the non-cognate ligand. Having validated the receptor models it should now be possible to search for or design, species-specific, molecules that may act as insecticides. Furthermore, the current results put into question those studies that indicate that the ACPR may be activated by AKHs or that the AKHR is activated by ACP.

## Figures and Tables

**Figure 1 biomolecules-14-00313-f001:**
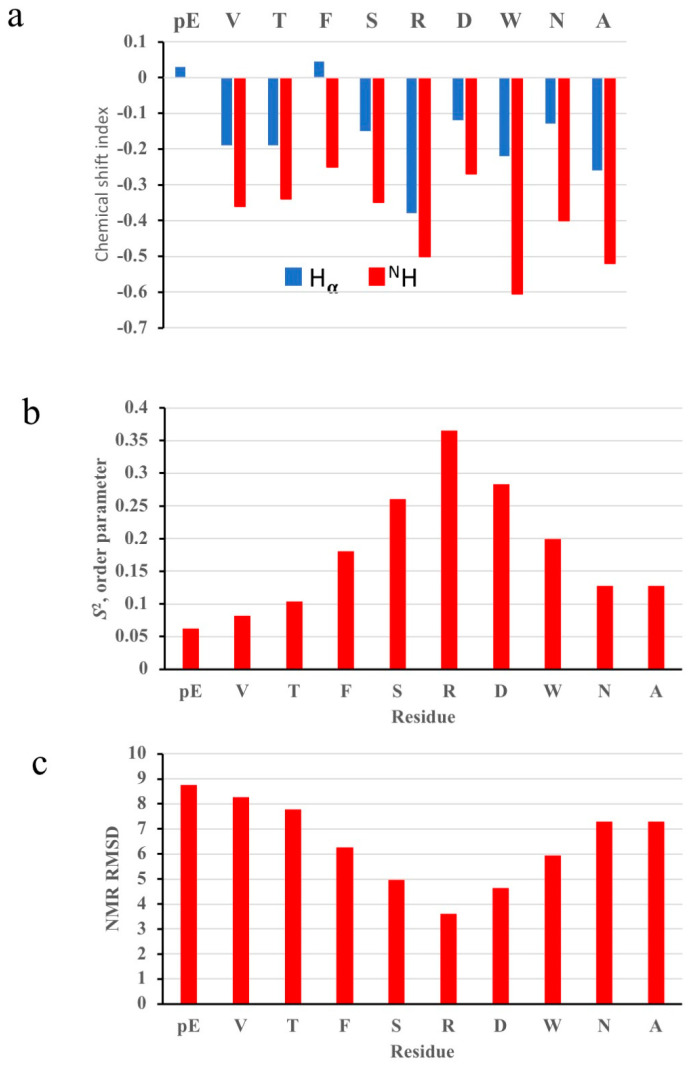
NMR results for Aedae-ACP in SDS micelle solution. (**a**) H_α_ and ^N^H random coil NMR chemical shift deviations; (**b**) *S*^2^ order parameter; and (**c**) NMR root mean square fluctuations.

**Figure 2 biomolecules-14-00313-f002:**
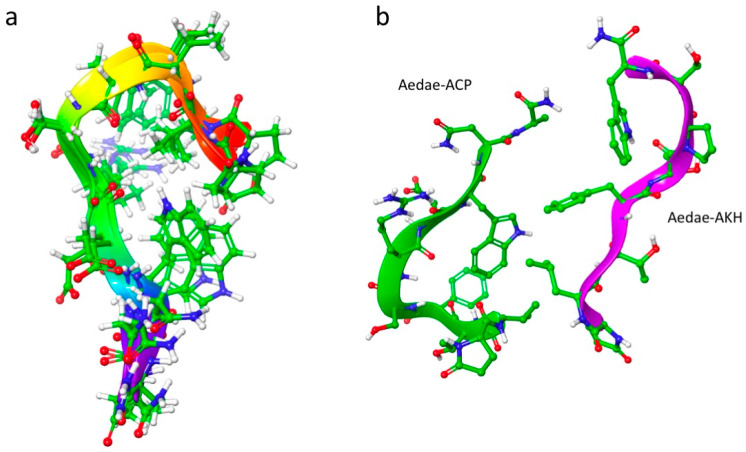
(**a**) Overlay of the largest cluster (49 members) from a 50 ns MD simulation of Aedae-ACP in DPC micelle solution. For clarity, every 10th snapshot is shown; (**b**) Comparison of Aedae-ACP and Aedae-AKH.

**Figure 3 biomolecules-14-00313-f003:**
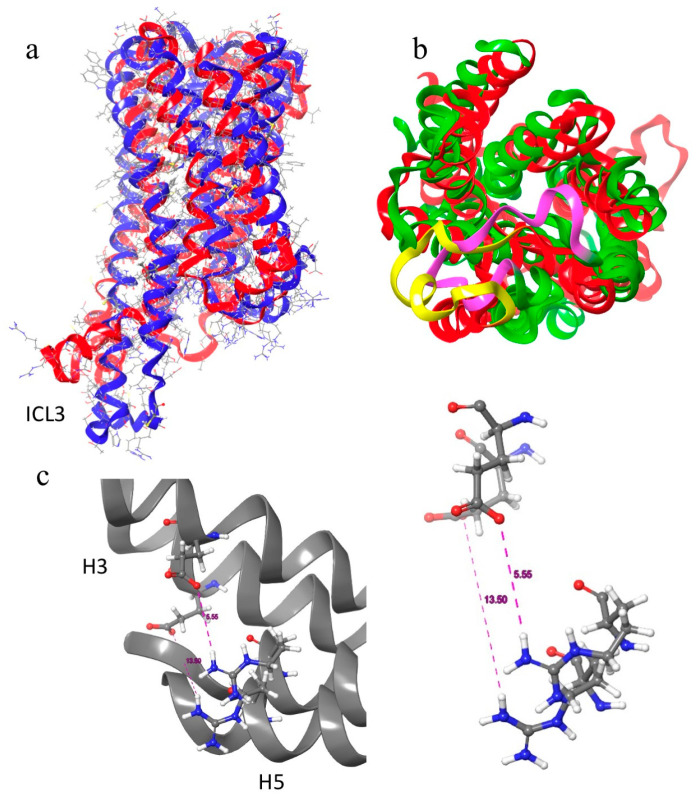
Overlay of two structures of Aedae-ACPR: (**a**) Side view, red = active state (6kr8); blue = inactive state (5D5A); (**b**) Top view highlighting the different positions of ECL2, yellow = inactive state; purple = active state; and (**c**) Ionic lock, distance between Glu^299^ and Arg^205^. Inactive (5D5A) 5.56 Å, active (6kr8) 13.5 A. Note how H3 and H5 have moved away from each other upon activation.

**Figure 4 biomolecules-14-00313-f004:**
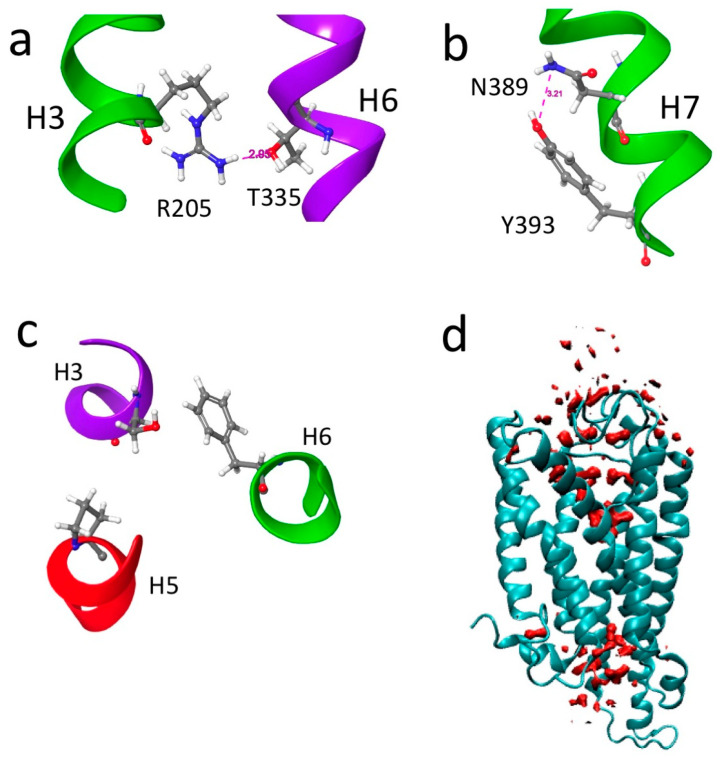
Aedae-ACP 6kr8 model: (**a**) DRC switch; (**b**) Tyrosine toggle switch; (**c**) hydrophobic connector switch; (**d**) Water density (above bulk) map of ACPR.

**Figure 5 biomolecules-14-00313-f005:**
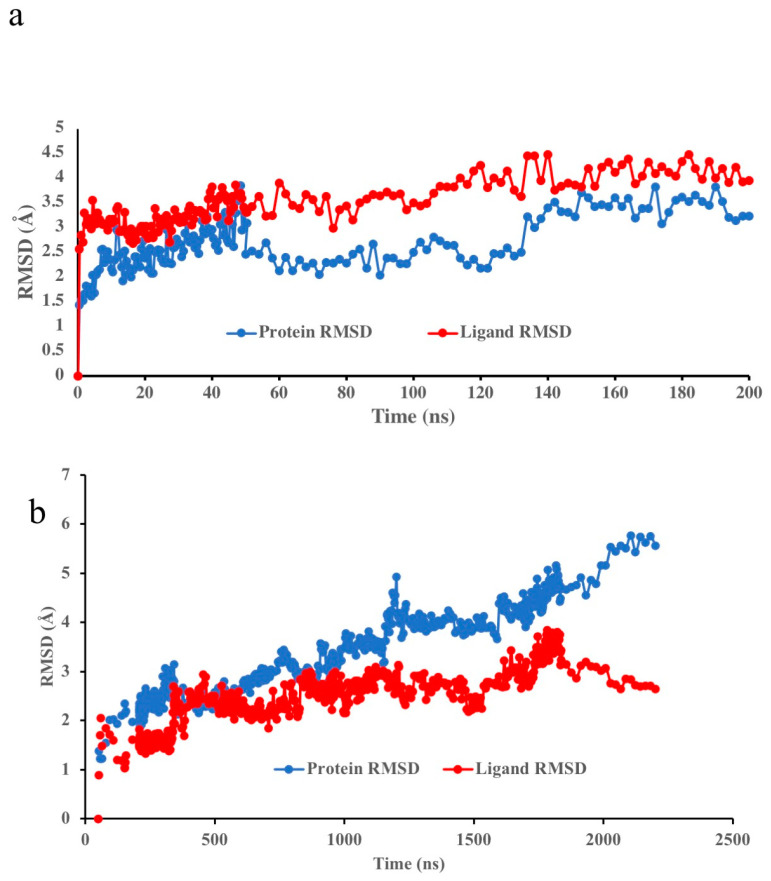
Protein RMSD during MD simulation of Aedae-ACP bound to Aedae-ACPR: (**a**) 6kr8 model; and (**b**) 5D5A model.

**Figure 6 biomolecules-14-00313-f006:**
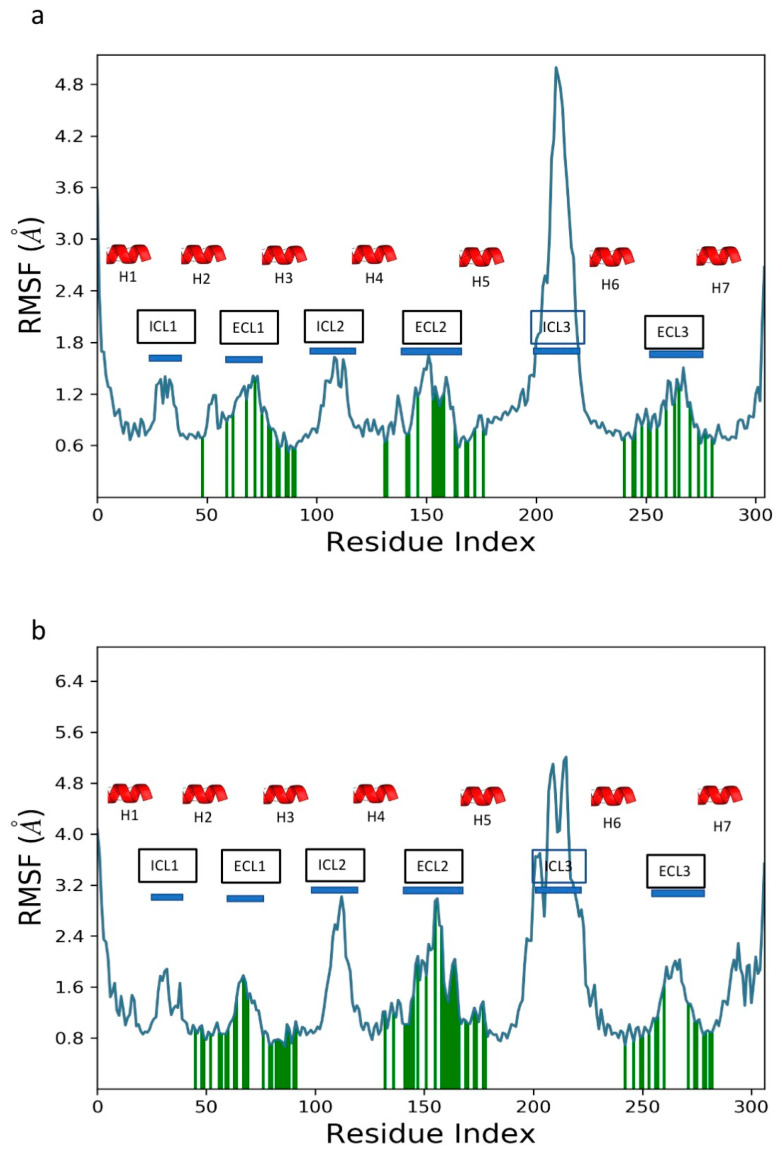
Protein RMSF of Aedae-ACPR + Aedae-ACP: (**a**) 6kr8 model; and (**b**) 5D5A model.

**Figure 7 biomolecules-14-00313-f007:**
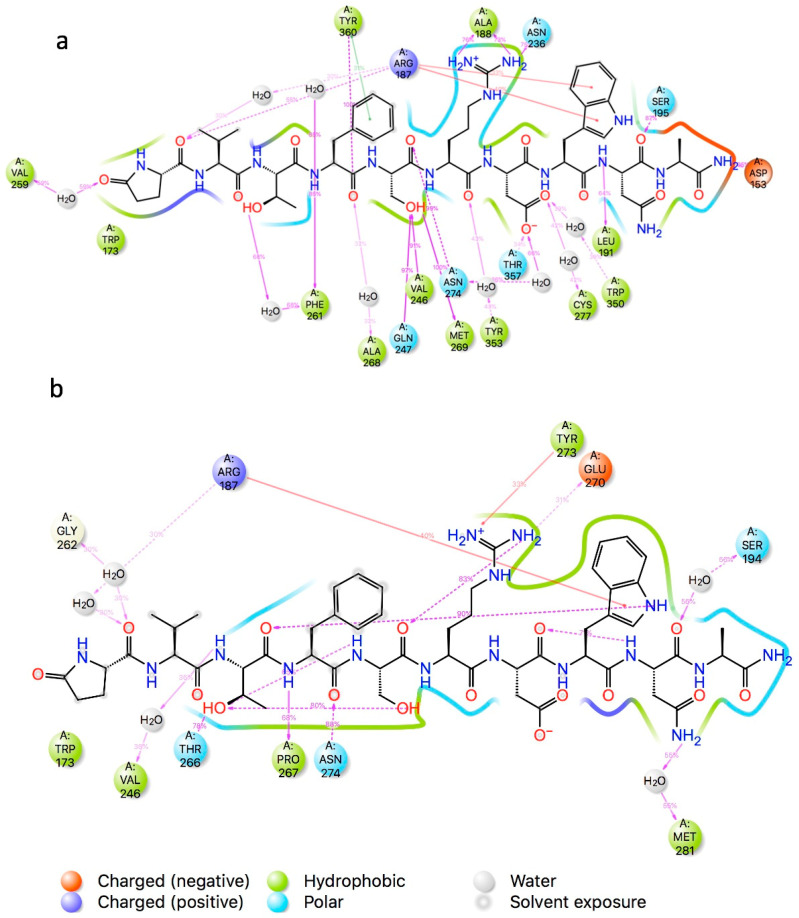
Simulation interaction diagram Aedae-ACPR + EVTFSRDWNAamide: (**a**) 6kr8 model; and (**b**) 5D5A model. A schematic of detailed ligand atom interactions with the protein residue interactions that occur more than 30.0% of the simulation time.

**Figure 8 biomolecules-14-00313-f008:**
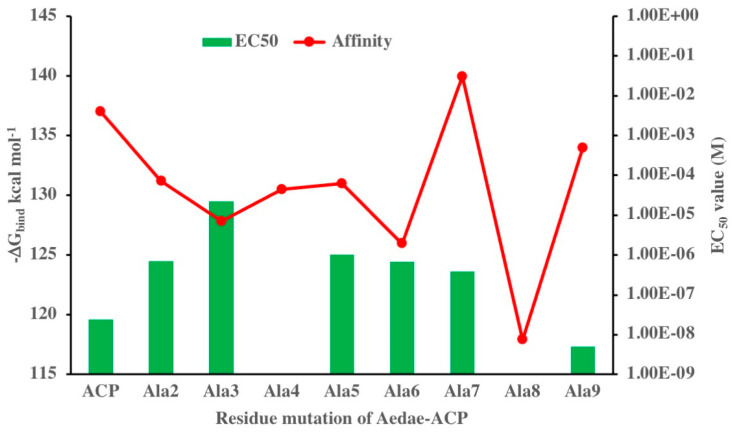
Free energy of binding and EC_50_ values for a series of alanine mutations of Aedae-ACP binding to Aedae-ACPR.

**Figure 9 biomolecules-14-00313-f009:**
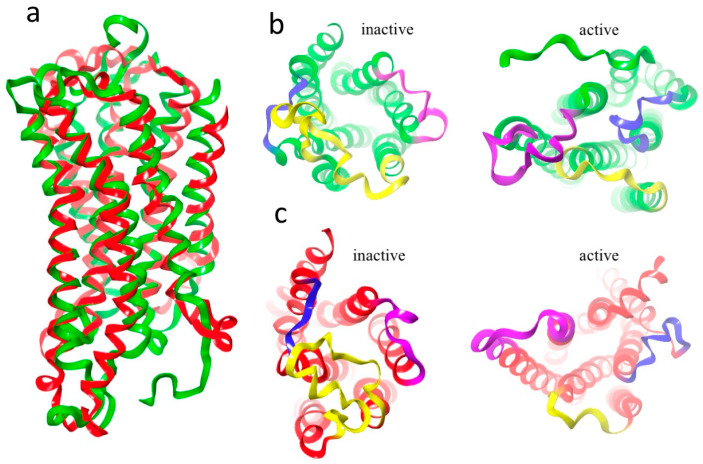
Aedae-AKHR: (**a**) Overlay of two models, inactive-green, active-red; (**b**) Top view showing extracellular loops blue = ECL1, yellow = ECL2, and purple = ECL3; and (**c**) Bottom view showing intracellular loops. Blue = ICL1, yellow = ICL2, plum = ICL3. For clarity, the clipping plan has been set so that only to top or bottom half of the receptor is shown.

**Figure 10 biomolecules-14-00313-f010:**
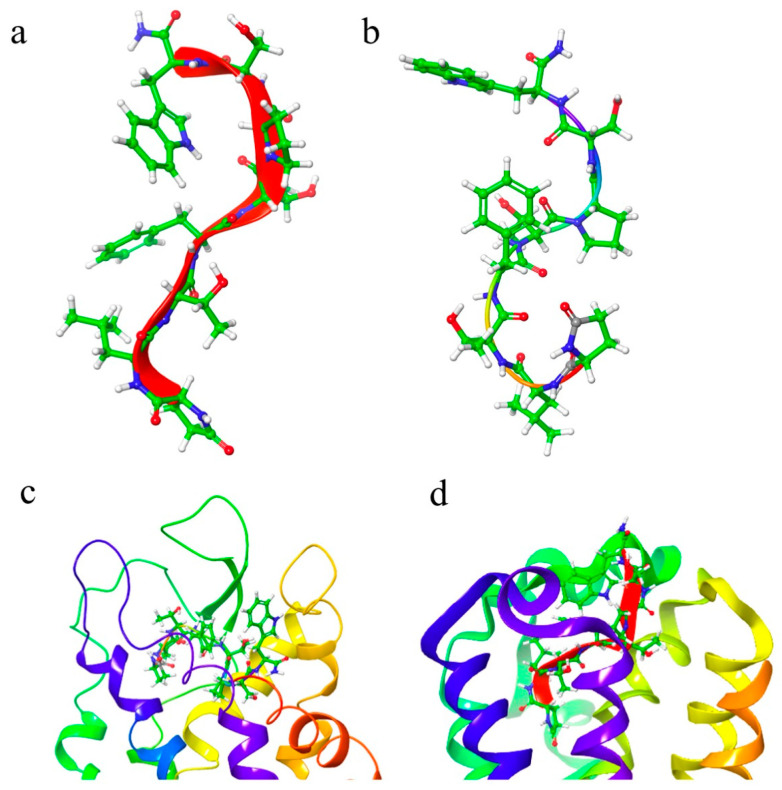
Conformation of Aedae-AKH bound to (**a**) Aedae-AKHR, and (**b**) Schgr-AKHR. The binding pocket of (**c**) Aedae-AKHR and (**d**) Schgr-AKHR shows the different orientations of the Aedae-AKH in the two receptors.

**Table 1 biomolecules-14-00313-t001:** NMR assignments for Aedae-ACP in SDS micelles ^a^.

Residue	N	H	HA	HB	HG	CA	CB	CG	Others
1 Glu	-	-	4.42	2.53, 1.96	2.36 *	59.7	28.4	32.3	
2 Val	120.0	7.96	4.11	1.97	0.88, 0.78	62.6	33.0	21.7, 21.2	
3 Thr	117.7	7.93	4.36	4.10	1.08	61.4	70.6	21.5	
4 Phe	122.3	8.02	4.70	3.19, 3.01		57.9	40.0		HD 7.24, HE 7.20, HZ 7.10, CD 131.8, CE 131.2, CZ 129.4
5 Ser	116.9	8.01	4.40	3.83, 3.75		58.5	64.3		
6 Arg	121.1	7.74	4.10	1.41, 1.52	1.33 *	55.9	30.7	27.3	NE 124.4, HD1 2.89, HD2 2.93, HE 6.87, CD 43.5
7 Asp	119.2	8.05	4.58	2.62, 2.50		-	39.2	-	
8 Trp	122.0	7.71	4.52	3.21, 3.26		57.9	29.7		NE 128.7, HD1 7.22, HE1 9.82, HE3 7.00, HZ2 7.37, HZ3 7.55, HH2 7.04, CD1 127.5, CE3 121.6, CZ2 114.5, CZ3 121.1, CH2 124.3
9 Asn	120.3	8.02	4.61	2.65, 2.48					ND2 112.4, HD2a 6.67, HD2b 7.32
10 Ala	124.0	7.71	4.13	1.31		52.8	19.6		

^a^ 2 mM peptide in 150 mM SDS micelles. Experiments were run at 310 K and referenced to TSP. * Identical chemical shift for both protons. - Not observed.

## Data Availability

Data are contained within the article and Appendix A.

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
