# Peer review of "The Adipokinetic Hormone (AKH) and the Adipokinetic Hormone/Corazonin-Related Peptide (ACP) Signalling Systems of the Yellow Fever Mosquito Aedes aegypti: Chemical Models of Binding"

_biomolecules, 2024, doi:10.3390/biom14030313_

Round 1

Reviewer 1 Report

Comments and Suggestions for Authors

The crucial roles in energy metabolism and balance are played by the insect AKH and ACP. The authors conducted NMR with MD calculations to explore the binding sites of these receptors and their docking models for ligands. The results indicate the evolutionary independence of the two systems, with receptors binding only to their cognate ligands. However, the paper falls short of meeting scientific writing standards, and I recommend reorganizing and resubmitting the manuscript.

In GPCR structure analysis, cryo-EM is generally considered a more reliable approach. Therefore, the necessity and advantages of using NMR in this study should be explicitly stated and emphasized. The original manuscript lacks figures, and tables are not presented in a standard format. The introduction, while detailed, is excessively redundant.

The "Materials and Methods" section lacks clarity, especially regarding the protein expression and purification methods, as well as the data processing and statistical procedures. The combination of "Results" and "Discussion" is not in accordance with the requirements of research papers in the Biomolecules Journal.

To address these issues, I recommend a thorough reorganization of the manuscript, clarifying the methodology, and presenting results and discussion in separate sections. Additionally, emphasis should be placed on justifying the use of NMR over cryo-EM and ensuring the inclusion of proper figures and tables in standard formats.

Comments on the Quality of English Language

Significant revisions in the English language are necessary.

Author Response

Response to the reviewers

Reviewer 1:

The crucial roles in energy metabolism and balance are played by the insect AKH and ACP. The authors conducted NMR with MD calculations to explore the binding sites of these receptors and their docking models for ligands. The results indicate the evolutionary independence of the two systems, with receptors binding only to their cognate ligands. However, the paper falls short of meeting scientific writing standards, and I recommend reorganizing and resubmitting the manuscript.

We thank the Reviewer for the time spent and the constructive comments. We have revised the manuscript in response to the comments and further clarified some points (see below). We hope to have improved the quality of our submission, already deemed “serious and well written” by Reviewer 2. 

In GPCR structure analysis, cryo-EM is generally considered a more reliable approach. Therefore, the necessity and advantages of using NMR in this study should be explicitly stated and emphasized.

The aims and experimental procedures used in our study do not involve the expression or structure determination of GPCRs. Rather, we have solved the structure of small peptides that are partially folded on a lipid membrane template, as has been previously published in this field. NMR is a more suitable structural technique compared to X-ray of cryo-EM for investigation of these molecular systems. We have added a sentence to clarify this L462-464.

The original manuscript lacks figures, and tables are not presented in a standard format.

The figures have been inserted near to their mention in the revised manuscript. The table has also been modified to the required format.

The introduction, while detailed, is excessively redundant.

Thank you for indicating. However, the Introduction fulfills the aim to make the reader familiar with the subject/topic of the study, which requires both a biological and chemical background. Since the intricacies of the study require both and because the study is published in a more biochemical journal it is prudent to introduce the topic from different angles. This we have attempted and could be why the Reviewer found it be “excessively redundant”. For most chemists or biologists, and possibly most of our readership, they may benefit from the “all inclusive” Introduction. Nevertheless, we have revised the Introduction slightly and omitted a few paragraphs which we deemed less important for understanding the background to the study. We are hopeful that the Reviewer will understand our motive and intention.

The "Materials and Methods" section lacks clarity, especially regarding the protein expression and purification methods, as well as the data processing and statistical procedures.

As stated above, expression, purification, and structure determinations of the GPCRs involved in this study were not performed. For statistical procedures, basic averages and routine standard deviations were performed and presented, which do not usually require their own M&M section.

The combination of "Results" and "Discussion" is not in accordance with the requirements of research papers in the Biomolecules Journal.

The instructions for authors stipulate that the discussion may be combined with results:

Discussion: Authors should discuss the results and how they can be interpreted in perspective of previous studies and of the working hypotheses. The findings and their implications should be discussed in the broadest context possible, and limitations of the work highlighted. Future research directions may also be mentioned. This section may be combined with Results.

To address these issues, I recommend a thorough reorganization of the manuscript, clarifying the methodology, and presenting results and discussion in separate sections. Additionally, emphasis should be placed on justifying the use of NMR over cryo-EM and ensuring the inclusion of proper figures and tables in standard formats.

Thank you for the comments. The revised manuscript has been provided with highlighted changes.

Reviewer 2 Report

Comments and Suggestions for Authors

Comments about the manuscript:

“The Adipokinetic Hormone (AKH) and the Adipokinetic Hormone/Corazonin-Related Peptide (ACP) Signalling Systems of the Yellow Fever Mosquito Aedes aegypti: Chemical Models of Binding”

In insects, there are three neuropeptide systems (adipokinetic hormone (AKH), corazonin (Crz), and adipokinetic hormone/peptide) and their receptors. The work presented here concerns the interactions of ligands with their receptors and, more particularly, the two signaling systems, AKH and ACP in the yellow fever mosquito, Aedes aegypti. To achieve this, the structure of the hormones was determined by a nuclear magnetic resonance method, followed by modeling of the two G protein-coupled receptors. The receptor binding sites were identified. The models were validated by comparing the computational results with the available experimental data, which made it possible to consider that the models were correct and could be used.

This complex work provides useful elements for understanding the ligand-peptide association at the molecular level. Not being a specialist in the techniques used in this study which I consider serious and well written, I will limit myself to a few minor remarks.

Page 2, lines 49-50. Write “(6-8)” instead of “(see reviews by Gäde and Auerswald, 2003(6); Gäde, 2004(7); Marco and Gäde, 2020(8)).”

Page 2, line 53. Write “(9, 10)” instead of “(see reviews by Kodrik, 2008(9); Kodrik et al., 2015(10)).”

Page 3, lines 120-122. Write “by measuring physiological actions in vivo (32, 33) or in a cellular mammalian expression system in vitro (34, 35).” instead of “by measuring physiological actions in vivo (see for example, Gäde and Hayes (32); Ziegler et al. (33)) or in a cellular mammalian expression system in vitro (see for example, Caers et al. (34); Marco et al. (35))”.

Page 3, line 137. Write “(37, 38)” instead of “s (see for example, Zubrzycki and Gäde, (37); Nair et al., (38))”.

Page 3, lines 138-140. Write ”with its receptor (39- 41).” Instead of “with its receptor (see for example, Mugumbate et al., (39); Jackson et al., (40); Abdulganiyyu et al., (41)).”

Page 8, figure 2: delete one of the two legends.

Author Response

Reviewer 2:

In insects, there are three neuropeptide systems (adipokinetic hormone (AKH), corazonin (Crz), and adipokinetic hormone/peptide) and their receptors. The work presented here concerns the interactions of ligands with their receptors and, more particularly, the two signaling systems, AKH and ACP in the yellow fever mosquito, Aedes aegypti. To achieve this, the structure of the hormones was determined by a nuclear magnetic resonance method, followed by modeling of the two G protein-coupled receptors. The receptor binding sites were identified. The models were validated by comparing the computational results with the available experimental data, which made it possible to consider that the models were correct and could be used.

This complex work provides useful elements for understanding the ligand-peptide association at the molecular level. Not being a specialist in the techniques used in this study which I consider serious and well written, I will limit myself to a few minor remarks.

We would like to thank the Reviewer for the kind comments and time spent on reading and providing suggestions to improve our manuscript. All the changes have been included in the revised manuscript as required.

Page 2, lines 49-50. Write “(6-8)” instead of “(see reviews by Gäde and Auerswald, 2003(6); Gäde, 2004(7); Marco and Gäde, 2020(8)).”

Page 2, line 53. Write “(9, 10)” instead of “(see reviews by Kodrik, 2008(9); Kodrik et al., 2015(10)).”

Page 3, lines 120-122. Write “by measuring physiological actions in vivo (32, 33) or in a cellular mammalian expression system in vitro (34, 35).” instead of “by measuring physiological actions in vivo (see for example, Gäde and Hayes (32); Ziegler et al. (33)) or in a cellular mammalian expression system in vitro (see for example, Caers et al. (34); Marco et al. (35))”.

Page 3, line 137. Write “(37, 38)” instead of “s (see for example, Zubrzycki and Gäde, (37); Nair et al., (38))”.

Page 3, lines 138-140. Write “with its receptor (39- 41).” Instead of “with its receptor (see for example, Mugumbate et al., (39); Jackson et al., (40); Abdulganiyyu et al., (41)).”

Page 8, figure 2: delete one of the two legends.

All changes have been made; see revised Introduction and Figure 2.

Round 2

Reviewer 1 Report

Comments and Suggestions for Authors

The authors responded to my comments and revised their manuscript accordingly.  And it could be considered for publication.